# Plasma midkine concentrations in healthy children, children with increased and decreased adiposity, and children with short stature

**Youn Hee Jee**[1⦿], **Kun Song Lee**[2⦿], **Shanna Yue**[1], **Ellen W. Leschek**[3], **Matthew G. Boden**[1], **Aysha Jadra**[1], **Anne Klibanski**[4], **Priya Vaidyanathan**[5], **Madhusmita Misra**[4], **Young Pyo Chang**[2], **Jack A. Yanovski**[1], **Jeffrey Baron**[1]*

**1** Eunice Kennedy Shriver National Institute of Child Health and Human Development, National Institutes of Health, Bethesda, MD, United States of America, **2** Pediatrics, Dankook University Hospital, Cheonan, Republic of Korea, **3** National Institute of Diabetes and Digestive and Kidney Diseases, National Institutes of Health, Bethesda, MD, United States of America, **4** Division of Pediatric Endocrinology, Massachusetts General Hospital for Children, Harvard Medical School, Boston, MA, United States of America, **5** Pediatric Endocrinology, Children's National Medical Center, Washington, DC, United States of America

⦿ These authors contributed equally to this work.
* jeffrey.baron@nih.gov

**Data Availability Statement:** All relevant data are within the manuscript and its Supporting Information files.

## Abstract

### Background

Midkine (MDK), one of the heparin-binding growth factors, is highly expressed in multiple organs during embryogenesis. Plasma concentrations have been reported to be elevated in patients with a variety of malignancies, in adults with obesity, and in children with short stature, diabetes, and obesity. However, the concentrations in healthy children and their relationships to age, nutrition, and linear growth have not been well studied.

### Subjects and methods

Plasma MDK was measured by immunoassay in 222 healthy, normal-weight children (age 0–18 yrs, 101 boys), 206 healthy adults (age 18–91 yrs, 60 males), 61 children with BMI ≥ 95th percentile (age 4–18 yrs, 20 boys), 20 girls and young women with anorexia nervosa (age 14–23 yrs), and 75 children with idiopathic short stature (age 3–18 yrs, 42 boys). Body fat was evaluated by dual-energy X-ray absorptiometry (DXA) in a subset of subjects. The associations of MDK with age, sex, adiposity, race/ethnicity and stature were evaluated.

### Results

In healthy children, plasma MDK concentrations declined with age (r = -0.54, *P* < 0.001) with values highest in infants. The decline occurred primarily during the first year of life. Plasma MDK did not significantly differ between males and females or between race/ethnic groups. MDK concentrations were not correlated with BMI SDS, fat mass (kg) or percent total body fat, and no difference in MDK was found between children with anorexia nervosa, healthy

**Funding:** This work was supported by the Intramural Research Program of the Eunice Kennedy Shriver National Institute of Child Health and Human Development (NICHD) grants Z1AHD00640 and Z1AHD00641 (to YHJ, SY, MGB, AJ, JAY, and JB), NIDDK (to EWL), the Thrasher Early Career Research Award ID#12555 and Pediatric Endocrine Society Research Fellowship Award (to YHJ), R01DK062249 (MM), and Dankook University Hospital Research Institute of Clinical Medicine grant 312-82-61231 (to KSL, YPC). The funders had no role in study design, data collection and analysis, decision to publish, or preparation of the manuscript.

**Competing interests:** The authors have declared that no competing interests exist.

weight and obesity. For children with idiopathic short stature, MDK concentrations did not differ significantly from normal height subjects, or according to height SDS or IGF-1 SDS.

## Conclusions

In healthy children, plasma MDK concentrations declined with age and were not significantly associated with sex, adiposity, or stature-for-age. These findings provide useful reference data for studies of plasma MDK in children with malignancies and other pathological conditions.

## Introduction

Midkine (MDK) is a heparin-binding growth factor that shares 45% amino acid sequence identity with the other heparin-binding growth factor, pleiotrophin [1]. MDK is rich in basic amino acid residues and binds to polysulfated glycosaminoglycans such as chondroitin sulfate and similarly has a high affinity for heparin [2]. Since its discovery, MDK has been found to have diverse activities, such as promoting cell growth and survival, cell migration, angiogenesis, fibrinolysis, and tissue repair. MDK is highly expressed in some malignancies [3–7] and is thought to play a role in oncogenesis. Elevated plasma or serum MDK concentrations have been reported in a wide variety of malignancies, including neuroblastoma [8], breast cancer [9], head and neck squamous cell carcinoma [10], hepatocellular carcinoma [11], and pediatric embryonal tumors [12]. Moreover, a recent study showed that midkine activates the mTOR pathway to induce neo-lymphangiogenesis, which supports metastatic spread of melanoma [7].

Although it has been widely studied in malignancies, the role of MDK in normal physiology has been investigated less extensively. MDK is highly expressed in multiple tissues of the embryo [13–14]. However, postnatally, MDK expression is downregulated in multiple organs of mice, rats, and sheep [15–16], resulting in low expression in adult tissues [13]. MDK knockout in mice has little phenotypic effect but double knockout of MDK and pleiotrophin impairs postnatal growth and causes infertility [17].

MDK has been studied in several endocrine-related conditions. MDK is expressed in adipocytes and regulated by inflammatory modulators, such as TNF-α and rosiglitazone [18]. Serum MDK concentrations were found to be elevated in adults with obesity [18]. Therefore, it has been suggested that MDK is a novel adipocyte-secreted factor associated with obesity [18]. In one prior study, children with a variety of endocrine conditions were studied as a control group for comparison to children with malignancies [19]. There appeared to be no association between age and serum MDK, but, unexpectedly, extremely high MDK concentrations were observed in occasional children with short stature, diabetes, and obesity [19]. In addition, MDK was reported to be lower in children treated with growth hormone injections. However, no follow-up study has been performed to confirm the findings in healthy children and children with obesity and growth disorders to date. Therefore, we designed a study to establish the reference ranges in healthy children and adults and investigated the association of MDK with different levels of adiposity and with disorders of childhood growth.

## Subjects and methods

### Study population

The study was approved by IRBs at the *Eunice Kennedy Shriver* National Institute of Child Health and Human Development, Dankook University Hospital, National Institute of

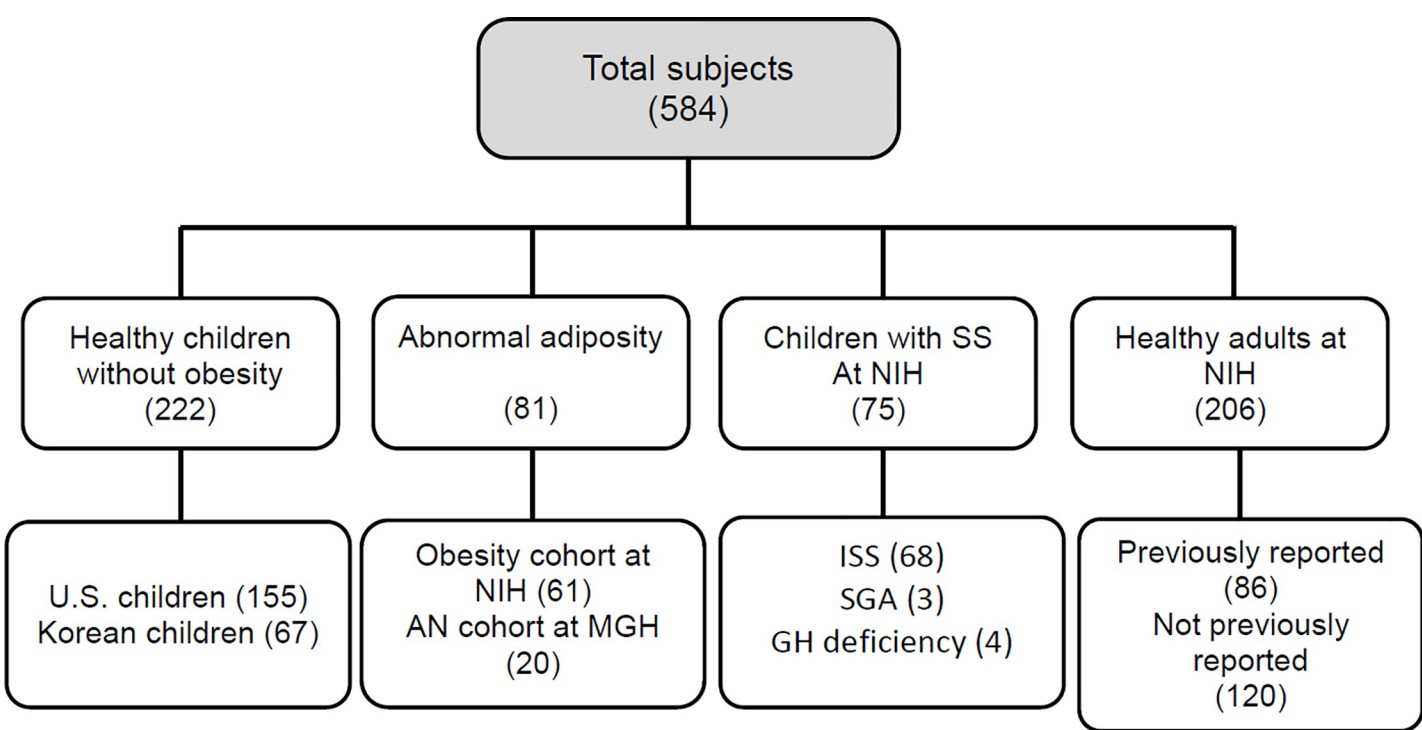

**Fig 1. Subject groups.** Healthy children without obesity, children and young adults with abnormal adiposity (obesity and anorexia nervosa), children with short stature, and healthy adults were studied. Numbers of subjects in each group are shown in parentheses. SS, short stature; AN, anorexia nervosa; ISS, idiopathic short stature; SGA, small for gestational age; GH, growth hormone; NIH, National Institutes of Health; MGH, Massachusetts General Hospital; U.S., United States (studied at NIH).

Diabetes and Digestive and Kidney, Massachusetts General Hospital, and Children's National Medical Center (ClinicalTrials.gov Identifiers: NCT00001195, NCT00001522, NCT00680979, NCT02311322). For all subjects, written informed consent and assent, if appropriate, were obtained. Written informed consent was obtained from all subjects and from parents or legal guardians of minors. Assent was also obtained from minors as appropriate. Subject characteristics are described in detail in **Fig 1** and **Table 1**.

**Healthy children.** Healthy children (n = 222 subjects, age 0–17.99 years, 101 male) were volunteers recruited under clinical protocols at the *Eunice Kennedy Shriver* National Institute of Child Health and Human Development (NICHD) in the United States (n = 155) and Dankook University Hospitals in South Korea (n = 67) between 2011 and 2016.

**Healthy adults.** Healthy adults (n = 206, age 18.2–91 yrs, male = 60) included 1) healthy, unaffected parents, siblings or relatives of children with growth disorders who participated in clinical studies at the NIH between 2011 and 2016, 2) healthy volunteers at the NIH (n = 120), and 3) otherwise healthy subjects with benign thyroid nodules (n = 86) [6].

**Children with obesity or anorexia nervosa (AN).** Otherwise healthy children with obesity (n = 61, BMI>95th percentile, age 4.2–17.9 yrs, male = 20) were recruited at the *Eunice Kennedy Shriver* National Institute of Child Health and Human Development (NICHD). Subjects with AN (n = 20, age 14–21 years, all female) were recruited at Massachusetts General Hospital, Boston, through eating disorder providers, local pediatricians, and treatment centers between 2011 and 2013 [20]. Inclusion criteria included a diagnosis of anorexia nervosa (AN) based on DSM-V criteria. The diagnosis of AN was confirmed by the study psychologist. Exclusion criteria included active suicidality, psychosis or substance abuse, and hematocrit

Table 1. Demographic and anthropometric data of subjects.

| | Healthy Children without Obesity (n = 222) | | Children with High and Low Adiposity | | Children with Short Stature (n = 75) | | Healthy Adults (n = 206) | |
| | | | Children with Obesity BMI ≥ 95 Percentile (n = 61) | | Children and adults with Anorexia Nervosa (n = 20) | | | |
|---|---|---|---|---|---|---|---|---|
| Median age [age range] (years) | 11.4 [0–17.9] | | 12.1 [4.2–17.9] | | 18.00 [14.0–23.0] | | 9.2 [3.6–17.4] | 41.2 [18.2–91.0] |
| Percent male (%) | 45.7 | | 32.8 | | 0 | | 55.00 | 29.6 |
| Race distribution (%) | White | 38.9 | White | 43.1 | White | 80.0 | White | 75.4 | White | 47.9 |
| | Non-Hispanic Black | 18.1 | Non-Hispanic Black | 37.9 | Non-Hispanic Black | 0 | Non-Hispanic Black | 6.6 | Non-Hispanic Black | 26.0 |
| | Asian | 35.7 | Asian | 6.9 | Asian | 10.0 | Asian | 8.2 | Asian | 7.8 |
| | Hispanic | 0 | Hispanic | 0 | Hispanic | 10.0 | Hispanic | 0 | Hispanic | 0 |
| | Other† | 7.3 | Other† | 12.1 | Other† | 0 | Other† | 9.8 | Other† | 18.2 |
| Median BMI SDS [range] | 0.49 [-2.24–1.62] | | 1.99 [1.64–3.14] | | -0.9 [-3.32—-0.31] | | -0.5 [-4.78–1.91] | 1.53 [-1.50–3.16] |
| Median height SDS [range] | 0.1 [-1.9–3.2] | | 0.75 [-1.8–3.1] | | | | -2.2 [-5.05—-0.8] * | -0.256 [-2.97–2.37] |

†: Other, more than one race/ethnicity

*: Including children on growth hormone treatment.

<30%, potassium <3.0 mmol/L, or glucose <50 mg/dL (to exclude subjects with severe acute illness). Use of psychiatric medications was not an exclusion criterion for study participation.

**Children with short stature.** Subjects (n = 68, age 0.6–17.4 yrs, male = 43) were children who visited the NIH Clinical Center pediatric endocrinology clinic for evaluation of short stature between 2011 and 2016. Inclusion criteria included height SDS < - 2.2 (at the time of presentation or before growth hormone initiation) without systemic illness or history of malignancy. An additional 3 subjects were born small for gestational and failed to undergo catch-up growth into the normal range.

Patients with growth hormone deficiency (n = 4) were diagnosed by two standard growth hormone provocative tests without estrogen priming. 3 subjects had a history of combined pituitary hormone deficiencies and 1 subject with isolated growth hormone deficiency who showed a peak growth hormone of 1 ng/mL. Patients who were receiving growth hormone for ISS and growth hormone deficiency were also included to study the impact of growth hormone on MDK concentrations.

**Measurements, Tanner staging, and body composition.** Subjects' weights were measured to the nearest 0.1 kg with a digital scale. Height was determined with a stadiometer from the average of three measurements recorded to the nearest millimeter. BMI z-scores were calculated according to the Centers for Disease Control and Prevention 2000 standards [21]. For 132 healthy children without obesity, Tanner staging was assessed by physical exam conducted by an experienced pediatric endocrine research nurse or pediatric endocrinologist. The Tanner stages were later obtained from the subjects' records. Breast development for girls and pubic hair for boys were used for analysis because these were the observations most widely recorded.

In 77 healthy subjects with normal weight (BMI SDS: -0.67 to 1.64, age 9.08–26 yrs) and 44 subjects with obesity (BMI SDS: 1.65 to 2.71, age 8.8–25.8 yrs)[22], total fat mass (kg) and body fat percent were measured by dual-energy X-ray absorptiometry (DXA) using a Hologic QDR Discovery instrument (Hologic, Bedford, MA).

**Blood collection.** Blood was collected from a peripheral vein in a plastic tube containing sodium citrate because glass tubes adsorb MDK [23]. Blood collection via heparinized catheter

was avoided [24]. The blood was centrifuged at 4˚C for 15 min at 3,000 g within 2 h of veni-puncture. Plasma was aliquoted in plastic tubes and stored at -80˚C until MDK assay.

## ELISA for plasma MDK

MDK was measured using a commercial sandwich enzyme-linked immunosorbent assay (ELISA) (BioVendor, US) with modifications to increase the sensitivity as previously described [6]. In summary, poly-L-lysine was added to the biotin-labelled detection antibody solution provided with the kit. For the assay, 125 µL of plasma were diluted in 125 µL of TBSTA (Tris-buffered saline, 1% bovine serum albumin, 0.5% tween 20, pH 7.4). The rest of the procedure was identical to the procedure previously described [6]. The detection limit for plasma MDK was 8.7 pg/mL. Intra-assay CV was 6% with 0.2 ng/mL and 3% with 0.8 ng/mL and inter-assay CV was 22%. The assay has been shown not to cross-react with pleiotrophin in a previous study [25].

## Statistical analysis

All MDK values were log-transformed to better approximate a normal distribution. The associations between MDK concentration and age, sex, race/ethnic group, height SDS and BMI SDS were assessed using ANOVA or Pearson correlation. Subjects < 2 years old were excluded from the BMI SDS analysis. Because MDK concentration correlated significantly only with age, differences in MDK concentrations between groups of subjects were adjusted for age, using age as a covariate in a general linear model in SPSS 19 (IBM, NY). MDK concentrations at different Tanner pubertal stages were compared using one-way ANOVA with Bonferroni correction for multiple post hoc comparisons. Data are presented as mean ± SEM, and a $P$ value of $\leq 0.05$ was considered statistically significant. In healthy children, the relationship between plasma MDK concentration and age was analyzed both by treating age as a continuous variable and also by categorizing for age: infancy (< 1 year), prepubertal age range (1–7 years), peripubertal to pubertal age range (8–14 years), and late pubertal to postpubertal age range (15–18 years).

## Results

In healthy children without obesity (n = 222), MDK decreased with age (r = -0.54, $P < 0.001$). The sharpest decline occurred during the first year of life (**Fig 2A and 2B**). MDK concentrations were not significantly associated with sex, race, BMI SDS or height SDS (**Table 2** and **Fig 2C and 2D**). MDK concentrations were also analyzed by pubertal stage in healthy children, age 8–18 years, who had Tanner staging performed at the time of blood collection (n = 132) (**Fig 2E**). The association between Tanner stage and MDK concentration did not reach statistical significance ($P = 0.27$) after adjusting for age. In healthy adults, MDK concentration was weakly positively correlated with age (r = 0.16, $P = 0.03$) but not with sex, race, height SDS or BMI SDS ($P = $ NS).

To investigate further the association between MDK concentrations and different levels of adiposity, MDK concentrations were compared between children and young adults with either normal weight, obesity, or anorexia nervosa. Plasma MDK showed no difference between subjects with normal BMI, obesity, and AN (0.21 ± 0.01 *vs*. 0.19 ± 0.06 *vs*. 0.19 ± 0.16 ng/mL, $P = $ NS, adjusted for age and sex). We further evaluated the association between MDK concentrations and adiposity by investigating healthy-weight individuals (n = 77, age range 9–26 years, BMI percentile 78.7 ± 15.8, mean ± SD) and subjects with obesity (n = 45, age 9–26 years, BMI percentile 97.3 ± 1.3) who had DXA scans at the time of blood collection. As expected, the mean of percent body fat by DXA was significantly different in healthy-weight individuals and

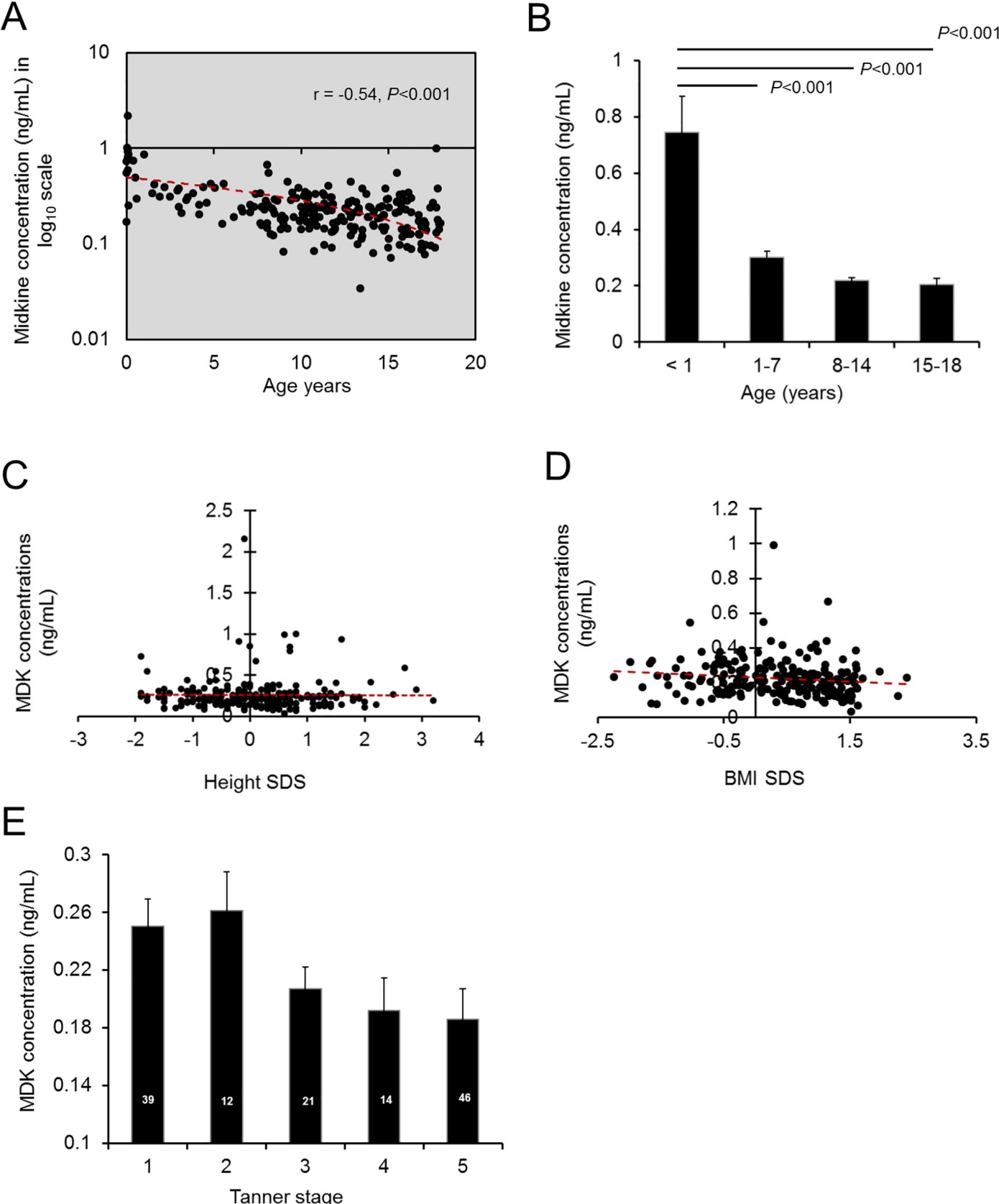

**Fig 2. Plasma midkine (MDK) concentrations in healthy children.** Plasma was obtained from healthy, non-obese children younger than 18 years (n = 222) from the United States or South Korea. Plasma midkine concentration was measured by sandwich enzyme-linked immunosorbent assay (ELISA). A) Scatterplot of plasma MDK concentrations (ng/mL) *vs* age. B) MDK concentrations (mean ± SEM) stratified by age. C) Scatterplot of plasma MDK concentrations *vs*. height standard deviation score (SDS) and D) Scatterplot of plasma MDK concentrations *vs*. body mass index (BMI) SDS. E) Plasma MDK concentrations (mean ± sem) stratified by Tanner stage in those subjects who had pubertal staging performed at the time of blood collection (n = 132).

subjects with obesity (30.7 ± 9.4% vs 41.2 ± 6.5, *P* < 0.001, adjusted for age and sex). In this combined group of subjects with and without obesity, the association, adjusted for age and sex, between MDK concentrations and percent body fat (%) or total fat mass in kg did not reach statistical significance.

To explore the relationship between MDK and linear growth, plasma MDK concentrations were measured in 68 children with idiopathic short stature (ISS, median height SDS -2.2) and compared to healthy children with stature within the normal range. The concentrations were similar in these two groups (0.24 ± 0.01 ng/mL *vs*. 0.26 ± 0.02 ng/mL, *P* = NS, adjusted for age and sex). MDK concentrations appeared similar in those children with idiopathic short stature who were receiving growth hormone treatment (n = 11, MDK 0.22 ± 0.03 ng/mL, mean ± SD), as well as in children receiving growth hormone for growth hormone deficiency (n = 4, MDK 0.26 ± 0.06 ng/mL) and children born SGA who failed to catch-up in to the normal range (n = 3, MDK 0.26 ± 0.06 ng/mL), although the small number of subjects in these categories precludes definitive conclusions. In children with idiopathic short stature, MDK was not significantly associated with height SDS, BMI SDS, or IGF-1 SDS (*P* = NS, adjusted for age and sex).

## Discussion

In healthy children and adults, we found that plasma MDK concentrations declined with age, with the steepest declines occurring in early childhood. Children in Tanner stage 1 showed higher MDK concentrations than in Tanner stage 5 but this effect appeared to be explained by their increasing age. Among healthy children, MDK concentrations did not correlate significantly with BMI SDS, total body fat mass, or percentage body fat. Moreover, MDK concentrations did not differ in children and young adults with anorexia nervosa or obesity compared to controls. In children with short stature, the MDK concentrations did not differ from normal children or show any association with height SDS, IGF-1 SDS, or among a small cohort receiving growth hormone treatment.

**Table 2. Association between plasma MDK concentrations and sex, race/ethnicity, BMI SDS and height SDS in healthy children.**

| Characteristic | N | Midkine concentration ng/mL, mean ± SEM | P value* |
|---|---|---|---|
| Sex | | | NS |
| Male | 101 | 0.28 ± 0.002 | |
| Female | 121 | 0.25 ± 0.02 | |
| Race/Ethnicity | | | NS |
| Asian | 79 | 0.35 ± 0.03 | |
| Non-Hispanic black | 40 | 0.18 ± 0.01 | |
| White | 87 | 0.23 ± 0.014 | |
| All other ethnicities | 16 | 0.2 ± 0.02 | |

N, number of subjects

*: analyzed by ANCOVA with age as a covariate

The observed decline in plasma MDK concentration with age, particularly in early childhood, is conceptually consistent with our previous studies showing that MDK concentrations in human amniotic fluid decline with gestational age [23] and that MDK mRNA expression decreases with age in multiple tissues of juvenile rats [15]. Taken together, these findings raise the possibility that high extracellular concentrations of MDK in early life may help support the rapid growth of infancy. Partially consistent with this hypothesis, double knockout of MDK and pleiotrophin, a related heparin-binding growth factor, impairs postnatal growth in mice [24], although knockout of MDK alone does not have this effect. The current finding that plasma MDK declines with age, particularly in early life, paralleling the decline in growth rate, is consistent with this hypothesis. Because of this possible role of MDK in normal childhood growth, we were particularly interested to see whether MDK might be altered in some children with growth disorders. Indeed, one prior study found that some children with short stature had extremely high circulating MDK concentrations [19]. However, we did not find abnormal plasma MDK concentrations in children with short stature.

Our findings do not confirm a reported association between circulating MDK concentrations and adiposity. Fan et al reported that serum MDK concentrations correlated with BMI in adults and that MDK concentrations were higher in adults with overweight and obesity than in those of normal weight [18]. We did not find an association between plasma MDK and BMI SDS, total body fat mass, or percent body fat. Furthermore, MDK concentrations were not significantly different in conditions involving extremes of adiposity–obesity or anorexia nervosa. The discrepancy between our findings and the prior report might be due to the age of the participants or to methodological differences. We measured MDK in plasma (with sodium citrate) rather than serum because serum collection tubes are coated with silicon and contain silica powder to enhance blood coagulation, which was previously reported to adsorb MDK [25]. We also collected all blood samples in plastic tubes because glass was found to adsorb MDK both in a prior study [23] and in our own pilot experiments. Furthermore, we did not use blood that was collected through a heparinized catheter because heparin was reported to increase MDK concentration [26]. It is also possible that the different antibodies used in different assays may detect different forms of MDK in the circulation. Our assay was designed to measure full-length MDK and does not cross react significantly with pleiotrophin, a related heparin-binding growth factor [6]. Methodological differences might also explain why we found circulating MDK concentrations to be approximately 10-fold lower than those reported by Fan et al [18].

Although MDK concentrations did not show a significant association with childhood growth disorders or obesity, the MDK concentration measured in healthy subjects might be useful as the reference range for patients with malignant disorders. Elevated plasma or serum MDK concentrations have been reported in a wide variety of malignancies [6, 8–12]. MDK expressed by malignancies may support lymphatic metastases by inducing neo-lymphangiogenesis, raising the possibility that plasma MDK may be useful for a biomarker in detection of invasive status of cancers.

In conclusion, we found that plasma MDK concentrations declined with age, particularly in early childhood, paralleling the normal decline in growth rate, suggesting that this heparin-binding growth factor might help support the rapid body growth of infancy. Plasma MDK was remarkably invariant with sex, adiposity, and in children with short stature.

## Supporting information

**S1 File. Subjects data.**
(XLSX)

## Author Contributions

**Conceptualization:** Youn Hee Jee, Jack A. Yanovski, Jeffrey Baron.

**Data curation:** Youn Hee Jee, Kun Song Lee, Shanna Yue, Ellen W. Leschek, Matthew G. Boden, Aysha Jadra, Anne Klibanski, Priya Vaidyanathan, Madhusmita Misra, Young Pyo Chang, Jack A. Yanovski.

**Formal analysis:** Youn Hee Jee, Ellen W. Leschek, Jack A. Yanovski.

**Funding acquisition:** Kun Song Lee, Ellen W. Leschek, Jeffrey Baron.

**Investigation:** Youn Hee Jee, Kun Song Lee, Young Pyo Chang, Jack A. Yanovski, Jeffrey Baron.

**Methodology:** Youn Hee Jee, Shanna Yue, Ellen W. Leschek, Matthew G. Boden, Anne Klibanski, Priya Vaidyanathan, Madhusmita Misra, Young Pyo Chang, Jack A. Yanovski.

**Project administration:** Jeffrey Baron.

**Supervision:** Jeffrey Baron.

**Validation:** Youn Hee Jee.

**Visualization:** Youn Hee Jee.

**Writing – original draft:** Youn Hee Jee.

**Writing – review & editing:** Youn Hee Jee, Kun Song Lee, Ellen W. Leschek, Madhusmita Misra, Jack A. Yanovski, Jeffrey Baron.

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
