## [Decision Letter · Decision Letter 0]

15 Jul 2019

PONE-D-19-15659

Plasma midkine concentrations in healthy children, children with increased and decreased adiposity, and children with short stature

PLOS ONE

Dear Dr Baron,

Thank you for submitting your manuscript to PLOS ONE. After careful consideration, we feel that it has merit but does not fully meet PLOS ONE’s publication criteria as it currently stands. Therefore, we invite you to submit a revised version of the manuscript that addresses the points raised during the review process.

You need to address comments # 1,2 and 4 by the reviewer and comments #1-4 from me.  Comment #3 from the reviewer and #5-6 from me are suggestions. 

We would appreciate receiving your revised manuscript by Aug 29 2019 11:59PM. To enhance the reproducibility of your results, we recommend that if applicable you deposit your laboratory protocols in protocols.io, where a protocol can be assigned its own identifier (DOI) such that it can be cited independently in the future. For instructions see: http://journals.plos.org/plosone/s/submission-guidelines#loc-laboratory-protocols

We look forward to receiving your revised manuscript.

Kind regards,

Kathleen E. Bethin, MD, PhD

Academic Editor

PLOS ONE

Journal Requirements:

2. You indicated that you had ethical approval for your study. In your Methods section, please ensure you have also stated whether you obtained consent from parents or guardians of the minors included in the study.

Additional Editor Comments:

1. The authors state,“All relevant data are within the manuscript and its Supporting Information files. “

However, the raw data are not included in the manuscript and there are no supporting files.

2. Figure 1 legend does not belong on the figure itself.

3. Figure 2 legend- must be reworked. A figure legend is not a description of the results. Please refer to reviewer comments.

4. For Figure 2 B please explain why/how the age brackets shown were chosen.

5. I agree with reviewer #1 assessment- The results section does not flow with the numbered sections. I would remove the numbers and the titles and add some transitions from one section to the other.

6.On page 11, lines 240-242 the MDK in children with ISS is compared to “healthy children.” This is implying that children with ISS are not healthy. I would suggest rewording the sentence.

.

Reviewers' comments:

Reviewer's Responses to Questions

**Comments to the Author**

1. Is the manuscript technically sound, and do the data support the conclusions?

Reviewer #1: Yes

2. Has the statistical analysis been performed appropriately and rigorously? 

Reviewer #1: Yes

3. Have the authors made all data underlying the findings in their manuscript fully available?

Reviewer #1: Yes

4. Is the manuscript presented in an intelligible fashion and written in standard English?

Reviewer #1: Yes

5. Review Comments to the Author

Reviewer #1: The submitted manuscript by Lee et al describes plasma midkine levels in different cohorts of children with normal weight, obesity and underweight, as well as short stature. This study was done because of prior observation that midkine levels were higher in obesity and short stature. The authors set out to clarify the midkine concentrations in these populations. The cohorts are well characterized and described. The data is clear and the manuscript is well-written. There a few items which could be addressed by the authors.

1) Page 7, line 133. The authors describe the Tanner staging for the pediatric cohort. However, they used different Tanner staging procedures based on sex -- females were staged for gonadarche, while males were staged by pubarche. The authors then evaluated midkine levels based on Tanner stage. However, the authors are combining 2 different pubertal assessments into one category and gonadarche is biochemically different than pubarche. The authors should discuss the rationale for this categorization. Alternatively, if they have data that would be consistent across both sexes, then midkine levels could be presented based on that Tanner staging.

2) Page 8, line 169. What does the term "parallelism" mean ? Do they mean to say correlation or consistency?

3) Results (starting on page 9) Do the individual results categories need to be numbered??

4) Figure legend for Figure 2 should be reworked. Would recommend as follows: This figure legend could be reworked

Fig 2A) Scatter-plot of log-transformed plasma MDK values (ng/ml) vs. age in healthy children (N= 222). B) MDK values (mean +/- sem) stratified by age (?? any stats for this, ie difference between <1 year and other age cohorts??) C &D) Scatterplots of plasma MDK concentrations vs. height (C) and BMI (D) (Show the r-value on the curves) E) MDK values (mean +/- sem) stratified by Tanner Stage.

6. PLOS authors have the option to publish the peer review history of their article (what does this mean?). If published, this will include your full peer review and any attached files.

Reviewer #1: No

---

## [Author Response · Author response to Decision Letter 0]

3 Sep 2019

August 5, 2016

Kathleen E. Bethin, MD, PhD

Academic Editor

PLOS ONE

Re: PONE-D-19-15659

Title: Plasma midkine concentrations in healthy children, children with increased and decreased adiposity, and children with short stature

Dear Dr. Bethin,

We would like to thank the reviewer and editor for their careful evaluation of our manuscript. As requested, we are providing the point-by-point response to the reviewer and editor and the revised version of our manuscript with track changes. The reviewer and editor made a number of important comments, which we carefully addressed, as detailed below. With these revisions, we believe that the paper is strengthened and therefore hope that the reviewer and editor will find that it is now suitable for publication in PLOS ONE. 

Sincerely,

Jeffrey Baron, MD

Chief, Section on Growth and Development

 

Reviewer comments:

Journal requirements #1: Please ensure that your manuscript meets PLOS ONE's style requirements, including those for file naming.

Response: We believe that our manuscript meets PLOS One’s style including the file 

 names. Please let us know if any adjustments are needed.

Journal requirements #2: In your Methods section, please ensure you have also stated whether 

 you obtained consent from parents or guardians of the minors included in the 

 study.

Response: The language of consent process was revised as below.

Page 4: Written informed consent was obtained from all subjects and from parents or 

legal guardians of minors. Assent was also obtained from minors as appropriate. 

Journal requirements #3: “Data not shown” in your manuscript does not meet our data sharing requirements. PLOS does not permit references to inaccessible data.

Response: The “data not shown” is not critical to the manuscript and therefore was 

 simply deleted.

Page 8: Plasma was obtained from subjects regardless of fasting or time of day because 

 these factors did not affect plasma MDK concentrations based on our evaluation 

 (data not shown). Measurement of MDK in serum did not show good parallelism 

 (data not shown) and the concentrations of MDK were far lower than in plasma, 

 therefore only plasma MDK concentrations are reported.

Editor comments #1: The authors state, “All relevant data are within the manuscript and its Supporting Information files.” However, the raw data are not included in the manuscript and there are no supporting files.

Response: The raw data in excel is now submitted as a supporting information file. 

Editor comments #2: Fig 1 legend does not belong on the figure itself.

 Response: The fig 1 legend is revised as below. 

Page 4: 

Fig 1. Subject groups. Healthy children without obesity, children and young adults with abnormal adiposity (obesity and anorexia nervosa), children with short stature, and healthy adults were studied. Numbers of subjects in each group are shown in parentheses. SS, short stature; AN, anorexia nervosa; ISS, idiopathic short stature; SGA, small for gestational age; GH, growth hormone; NIH, National Institutes of Health; MGH, Massachusetts General Hospital; U.S., United States (studied at NIH)

Editor comment #3: Fig 2 legend- must be reworked. A figure legend is not a description of the results. Please refer to reviewer comments.

 Response: Thank you for the editor’s and reviewer’s comment. Fig 2 legend is revised 

 as below. 

 Page 10: 

Fig 2. Plasma midkine (MDK) concentrations in healthy children. Plasma was obtained from healthy, non-obese children younger than 18 years (n=222) from the United States or South Korea. Plasma MDK concentration was measured by sandwich enzyme-linked immunosorbent assay (ELISA). A) Scatterplot of plasma midkine (MDK) concentrations (ng/mL) vs age. B) MDK concentrations (mean ± SEM) stratified by age. C) Scatterplot of plasma MDK concentrations vs. height standard deviation score (SDS) and D) Scatterplot of plasma MDK concentrations vs. body mass index (BMI) SDS. E) Plasma MDK concentrations (mean ± sem) stratified by Tanner stage in those subjects who had pubertal staging performed at the time of blood collection (n = 132). 

Editor comment #4: For Fig 2 B please explain why/how the age brackets shown were chosen.

Response: The following sentence was added to the methods section:

Page9: 

In healthy children, the relationship between plasma MDK concentration and age was analyzed both by treating age as a continuous variable and also by categorizing for age: infancy (< 1 year), prepubertal age range (1-7 years), peripubertal to pubertal age range (8-14 years), and late pubertal to postpubertal age range (15-18 years). 

Editor comment #5: I agree with reviewer #1 assessment- The results section does not flow with the numbered sections. I would remove the numbers and the titles and add some transitions from one section to the other.

 Response: As suggested, the numbers and the titles in result section were removed and 

 the transitions were revised. 

Editor comment #6: On page 11, lines 240-242 the MDK in children with ISS is compared to “healthy children.” This is implying that children with ISS are not healthy. I would suggest rewording the sentence.

 Response: Thank you for the editor’s comment. We revised the sentence as below. 

Page 11: To explore the relationship between MDK and linear growth, plasma midkine concentrations were measured in 68 children with idiopathic short stature (ISS, median height SDS -2.2) and compared to healthy children with stature within the normal range. 

Lastly, we would like to inform the editor and reviewer that we revised Fig 1 for Y-axis to contain raw data in a log scale (not log-transformed data) to help readers.

Reviewer #1: The submitted manuscript by Lee et al describes plasma midkine levels in different cohorts of children with normal weight, obesity and underweight, as well as short stature. This study was done because of prior observation that midkine levels were higher in obesity and short stature. The authors set out to clarify the midkine concentrations in these populations. The cohorts are well characterized and described. The data is clear and the manuscript is well-written. There a few items which could be addressed by the authors.

1) Page 7, line 133. The authors describe the Tanner staging for the pediatric cohort. However, they used different Tanner staging procedures based on sex -- females were staged for gonadarche, while males were staged by pubarche. The authors then evaluated midkine levels based on Tanner stage. However, the authors are combining 2 different pubertal assessments into one category and gonadarche is biochemically different than pubarche. The authors should discuss the rationale for this categorization. Alternatively, if they have data that would be consistent across both sexes, then midkine levels could be presented based on that Tanner staging.

Response: We agree with the reviewer’s point. The data were obtained from subject’s medical record at the time of blood collection and breast development for girls and pubic hair for boys were primarily available. To address the reviewer’s point, we revised our manuscript as follows. 

Page 8:

For 131 healthy children without obesity, Tanner staging was assessed by physical exam conducted by an experienced pediatric endocrine research nurse or pediatric endocrinologist. The Tanner stages were later obtained from the subjects’ records. Breast development for girls and pubic hair for boys were used for analysis because these were the observations most widely recorded. 

2) Page 8, line 169. What does the term "parallelism" mean ? Do they mean to say correlation or consistency?

Response: We used the term “parallelism” to refer to a desirable property of an immunoassay. To assess parallelism, a subject’s plasma with high endogenous MDK is serially diluted and then assayed. If the measured concentrations of MDK decrease proportionally to the dilution factor, then the assay is said to be parallel, indicating that the binding characteristic of the antibodies to endogenous MDK is the same as antibody binding to the calibrator MDK and that the blood plasma does not create a matrix effect which alters the binding. However, in the revised manuscript, we decided not to show the parallelism data because it is not critical to the study, and we therefore deleted the sentence referring to parallelism. 

3) Results (starting on page 9) Do the individual results categories need to be numbered??

Response: We agree that the numbering is not necessary, and therefore we removed the numbers and subtitles and instead used the first sentence of each paragraph to aid the reader with the transitions. 

4) Figure legend for Fig 2 should be reworked. Would recommend as follows: This figure legend could be reworked

 Response: We appreciate the reviewer’s comment and revised the legend as suggested.

Fig 2. Plasma midkine concentrations in healthy children. Plasma was obtained from healthy, non-obese children younger than 18 years (n=222) from the United States or South Korea. Plasma midkine concentration was measured by sandwich enzyme-linked immunosorbent assay (ELISA) A) Scatterplot of plasma midkine (MDK) concentrations (ng/mL) vs age. B) MDK concentrations (mean ± SEM) stratified by age. C) Scatterplot of plasma MDK concentrations vs. height standard deviation score (SDS) and D) Scatterplot of plasma MDK concentrations vs. body mass index (BMI) SDS. E) Plasma MDK concentrations (mean ± sem) stratified by Tanner stage in those subjects who had pubertal staging performed at the time of blood collection (n = 132).

---

## [Editor Report · Decision Letter 1]

9 Sep 2019

PONE-D-19-15659R1

Plasma midkine concentrations in healthy children, children with increased and decreased adiposity, and children with short stature

PLOS ONE

Dear Dr Baron,

Thank you for submitting your manuscript to PLOS ONE. After careful consideration, we feel that it has merit but does not fully meet PLOS ONE’s publication criteria as it currently stands. Therefore, we invite you to submit a revised version of the manuscript that addresses the points raised during the review process.

We would appreciate receiving your revised manuscript by Oct 24 2019 11:59PM. To enhance the reproducibility of your results, we recommend that if applicable you deposit your laboratory protocols in protocols.io, where a protocol can be assigned its own identifier (DOI) such that it can be cited independently in the future. For instructions see: http://journals.plos.org/plosone/s/submission-guidelines#loc-laboratory-protocols

We look forward to receiving your revised manuscript.

Kind regards,

Kathleen E. Bethin, MD, PhD

Academic Editor

PLOS ONE

Additional Editor Comments (if provided):

All of the reviewers/ editors comments were addressed. However, 2 minor issues were discovered on this review:

1. The number of healthy children with Tanner staging is listed as 131 on lines 155-159 of the text and listed as 132 in figure 2.

2. The term "non-Hispanic Black" was used in Table 1 and "Africa American" was used in table 2. Please be consistent. Also, legend for Table 2 defines AA as African American but AA is not used in the Table.

---

## [Author Response · Author response to Decision Letter 1]

25 Sep 2019

Reviewer’s comments:

1. The number of healthy children with Tanner staging is listed as 131 on lines 155-159 of the text and listed as 132 in figure 2.

Response: We appreciate the reviewer’s careful review. We confirmed that 132 in Figure 2 is the correct number. Therefore, the text was revised to 132. 

2. The term "non-Hispanic Black" was used in Table 1 and "Africa American" was used in table 2. Please be consistent. Also, legend for Table 2 defines AA as African American but AA is not used in the Table.

Response: We revised our Tables to be consistent with non-Hispanic black and removed African American and AA.

---

## [Editor Report · Decision Letter 2]

7 Oct 2019

Plasma midkine concentrations in healthy children, children with increased and decreased adiposity, and children with short stature

PONE-D-19-15659R2

Dear Dr. Baron,

We are pleased to inform you that your manuscript has been judged scientifically suitable for publication and will be formally accepted for publication once it complies with all outstanding technical requirements.

With kind regards,

Kathleen E. Bethin, MD, PhD

Academic Editor

PLOS ONE
---

## [Editor Report · Acceptance letter]

18 Oct 2019

PONE-D-19-15659R2 

Plasma midkine concentrations in healthy children, children with increased and decreased adiposity, and children with short stature 

Dear Dr. Baron:

I am pleased to inform you that your manuscript has been deemed suitable for publication in PLOS ONE. Congratulations! Your manuscript is now with our production department. 

With kind regards,

on behalf of

Dr. Kathleen E. Bethin 

Academic Editor

PLOS ONE